# Markusen's Typology with a "European" Twist, the Examples of the French Aerospace Valley Cluster and the Andalucia Aerospace Cluster

**Vasileios Kyriazis and Theodore Metaxas ***

Department of Economics, University of Thessaly, 13451 Athens, Greece
* Correspondence: metaxas@uth.gr

**Abstract:** The phenomenon of firms grouping together has been extensively researched and is commonly known as industrial clusters. There are various ways to categorize these clusters, and in this paper, we adopt Markusen's classification, which identifies four distinct types of industrial districts: the Marshallian/Italianate type, the hub-and-spoke type, the satellite industrial platforms, and the state-anchored clusters. Adding to Markusen's typology, we will also try to delineate these two clusters' "European Aspects". We will examine if they have developed any "inter-European" synergy/ies with other entities (clusters, companies, E.U. institutions, etc.) of the E.U. ecosystem. The creation of such synergies includes the creation of technology innovation and interpersonal networks to serve as conduits for the diffusion of knowledge and exchange of information, the development of innovation initiatives between the entities of the technological ecosystem of the E.U. defense industry, and the creation of tangible "knowledge links". The aim of this study is to investigate which of the four types of industrial clusters described by Markusen the French Aerospace Valley cluster of the Midi-Pyrénées and Aquitaine regions and the Andalucia Aerospace cluster belong to.

**Keywords:** Markusen's typology; industrial clusters; aerospace clusters; comparative analysis

## 1. Introduction: Industrial Clusters, Geography, and Institution Building

Changes in modern economics and technology development have diminished many of the traditional roles of location. Globalization and the new business ecosystem it creates brought dramatic changes in terms of space, time, and practices. For example, removing bureaucratic formalities and procedures helps companies source capital, goods, information, and technology from around the world, often in a very easy manner, diminishing the role of location in competition [1]. Paradoxically, clusters reverse this general assumption. To put it simply, if geography plays a less significant role, why are the odds of finding a high-performance auto company in southern Germany or a fashion shoe company in northern Italy higher than in most other places [1]? The reason is that today's globalized economy may diminish the value of geography, but the geographic concentration is still valuable [2].

The fact that geography still has a role in analyzing financial phenomena resurged the cluster concept as one of the critical issues in the research agendas of scholars, "portraiting" it as a subject of intense research studies and economic analysis. As a result, some different definitions and typologies by which clusters are analyzed and categorized [3–8] have been developed.

Nevertheless, these definitions and analytical frameworks are conflictual. This is why it has become almost a common practice to begin any study/discussion on economic clusters with the "disclaimer" that there is no adequate, universally accepted definition of the phenomenon [9]. As Martin and Sunley [10] say, constructing a critical and solid review of clusters is a difficult task because there are many different varieties and types of clusters

and a constant feeling that there must be "more on it than this," creating a misbelief of a chaotic concept and/or a policy panacea.

Clusters expose quite specific and distinct features in many cases, which can be summarized as follows. An Industry A cluster refers to a collection of companies that share access to local resources, utilize comparable technologies, and establish connections and partnerships (alliances) with each other. These linkages can take the form of buyer–supplier relationships, sharing of human resources, machinery and/or infrastructure, joint marketing, training, or research initiatives, associations, and lobbying [1]. Businesses and institutions engage with one another at various levels within the ecosystem of a cluster. Engagement allows individual companies to increase their competitive advantage [11–13] through the creation of business synergies [14] and the pooling of resources, knowledge, and innovation [15,16]. Hence, an industrial cluster may be seen as an initiative to organize the participating members in a coordinated manner, where local rivalry/competition is used creatively to generate innovation to increase competitiveness by facilitating cooperation between companies, companies and R&D agencies/institutions, and between companies and local, regional, and/or national government [17].

A critical feature for analysis whenever a cluster is to be defined/set up is the possible linkages between firms and the implications these may have on "shared" strategies. This is a situation in which companies may compete in some respects (for example, as far as output markets) yet cooperate in other directions (for example, as far as joint training programs). Beyond this oxymoron, it is equally important to denote that a core prerequisite for the success of any cluster is the ability to bring together private and public stakeholders in a collaborative "working group" structure aiming to develop the associated cluster strategies. Collective problem solving and open-information sharing are critical factors in developing and effectively implementing regional competitiveness strategies for the clusters' development.

Once a cluster is defined and set up according to the features above and prerequisites, the implementation phase can follow several actions from the companies and the regional/national authorities. For example, a cluster may form around a sizeable competitive firm. Boeing serves as the central hub for the aerospace industry in Seattle, while Microsoft plays a similar role for the software industry. Similarly, the Fred Hutchinson Cancer Center and the University of Washington have played a significant role in shaping the development and organization of the biotechnology industry in the local area [18]. Additionally, the presence and support of a significant research institution may help and facilitate the development of a cluster, such as the information technology cluster in Silicon Valley (attributed to the research results and initiatives of Stanford University). The development of industry clusters can also be facilitated by the presence of specialized infrastructure conditions or resources. Clusters may include government, non-profit organizations, educational institutions, and other infrastructure and service providers whose presence is key to the cluster's strength.

The definition of a cluster in this study is not limited to the physical proximity of firms and institutions but also considers the nature and intensity of their interactions and relationships and the specific characteristics of their industry. Therefore, the definition goes beyond the traditional concept of the physical proximity of firms and institutions. While physical proximity is an essential factor that enables face-to-face interactions and knowledge exchange, the definition of a cluster in this study also considers the nature and intensity of firms' interactions and relationships.

Therefore, the definition provides a nuanced and multidimensional concept that considers the nature and intensity of the interactions and relationships between firms and institutions, the specific characteristics of their industry or field, as well as the E.U. institutional and policy framework that supports and shapes their development. This multidimensional concept of a cluster allows for a more comprehensive analysis of the factors that contribute to the success and competitiveness of industrial clusters and provides

policymakers and industry leaders with a more nuanced understanding of the strategies and policies needed to support their growth and innovation.

The definition creates a de facto exciting research aspect, reinforcing the article's scientific novelty. This lies in its "European" twist, which refers to the articles' focus on the specificities of inter-European collaboration between industrial clusters. By analyzing these two aerospace and defense clusters in France and Spain, the paper provides insights into the unique features and challenges of industrial clusters in Europe and the opportunities for cross-border collaboration and knowledge sharing.

Moreover, the article highlights the importance of European Union (E.U.) policies and initiatives in promoting the development of industrial clusters and innovation ecosystems. It argues that E.U. policies, such as the Horizon 2020 research and innovation program, can facilitate cross-border collaboration and knowledge exchange and provide financial support for developing innovative projects and initiatives, creating synergies and strategic alliances between different actors of the E.U. clusters' ecosystem.

Finally, the study aims to highlight the role of policy interventions in the development of industrial clusters in Europe, particularly in fostering innovation and knowledge creation, supporting the development of small and medium-sized enterprises, and promoting collaboration between industry, academia, and government. Overall, the study aims to provide insights into the unique features and challenges of industrial clusters in Europe and the opportunities for cross-border collaboration and knowledge sharing.

The paper will achieve this by investigating which of the four types of industrial clusters described by Markusen the French Aerospace Valley cluster of the Midi-Pyrénées and Aquitaine regions and the Andalucia Aerospace cluster belong to. Adding to Markusen's typology, we will also try to delineate these two clusters' "European Aspects". We will examine if they have developed any "inter-European" synergy/ies with other entities (clusters, companies, E.U. institutions, etc.) of the E.U. ecosystem.

The study aims to provide insights and policy implications for policymakers and industry leaders interested in supporting their growth and competitiveness. Overall, the study aims to contribute to the ongoing conversation on how to support the growth and competitiveness of industrial clusters in Europe and beyond and to provide policymakers and industry leaders with valuable insights and policy implications for achieving this goal, by also using E.U. "tools" and financial frameworks.

## 2. Cluster Prerequisites; or What the Main "Ingredients of a Cluster" Are

From those mentioned earlier, one can understand that industry, innovation, and governmental institutions are some of the main "ingredients" for forming a cluster.

**Industry**: For any cluster to develop, a large concentration of interconnected companies (either "vertically" or "horizontally" or in terms of location) is needed [19]. These companies could be dispersed over a geographical region but operate in a common or closely related business sector. For small and medium-sized enterprises (SMEs), being part of a cluster and engaging with competitors and established players from associated industries can enhance their competitiveness, resulting in faster growth and increased market recognition and status [20,21]. However, the structure of clusters can sometimes be unsymmetrical and hierarchical [22], with some companies having greater financial and institutional weight and acting as a hub, which can shape the development and structure of the cluster [23]. These companies may be located within the location of the cluster or elsewhere. Sometimes, the "hub" companies may not be based within the cluster's geographical boundaries. Multinational enterprises are often attracted to clusters once clusters are recognized as "experts" in the related industries. In fact, the inclusion of renowned foreign-owned companies in a cluster could further enhance its leadership in the related business directions and contribute to its business success, according to research by Julian Brikinshaw [24].

Innovation: A number of scholars have addressed the necessity of innovation and innovative products as an essential prerequisite for a cluster. In fact, it has been pointed out that

innovation may be a more critical success factor than low-cost production [25]. However, research and development on innovative products can be costly, and not many companies can assume the related costs. Yet, industrial clusters can, by definition, share the production and research and development costs for any new/innovative product/service. Available data show cluster initiatives are more likely to produce innovative products/services even with limited research and development resources. The E.U. cluster authority (INNOVA) has published its survey results, proving that for the vast majority of regions that host clusters, a significant improvement in the development of innovative products and services was demonstrated [26]. Specifically, the survey noted that at the European level, innovative cluster firms:

- Are more innovative than non-cluster firms: Over a specific period, 78% of cluster firms introduced significantly improved products/services, compared to 74% of non-cluster firms, during the same period. Similarly, 63% of innovative cluster firms introduced innovative production technology compared to 56% of non-cluster firms. In other words, innovation is spurred by such cluster initiatives where the "cross-pollination" of ideas occurs.
- Are more than twice as likely to source out research to other firms, universities, or public labs compared to the average non-cluster innovative European firms. This supports the view that clusters encourage knowledge dissemination/sharing, which stimulates innovation through cross-conception and exchange of ideas.
- Patent and Trademark their innovations more often: Over the same survey period, when only 12% of innovative non-cluster companies applied for a patent, the percentage of cluster companies that did the same was 29%.

These, as well as some further related findings of the study, are illustrated in the following Figure 1.

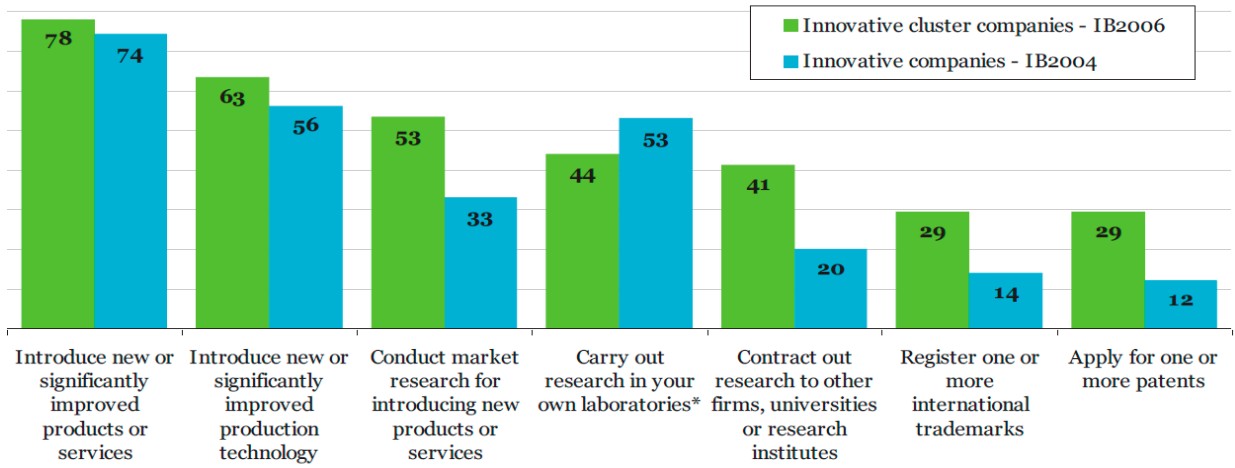

**Figure 1.** Comparison between Clustered and non-Clustered firms. Source: "The Concept of Clusters and Cluster Policies and their role for Competitiveness and Innovation: Main statistical results and lessons learned".

An important item to note, concerning the above figure, is the aspect "Carry out research in your own laboratories" [27]. Evidently, cluster companies have not developed their individual research and development capabilities but instead have increased the generation of innovative products/services through collaboration. This observation is further corroborated by the results in the item "Contract out research to other firms, universities or research institutes". As a result, it can be deduced that clusters typically promote innovation through collaborative and/or outsourced research and development.

A problem when evaluating the impact of clustering on innovation concerning local economies is the lack of adequate statistical data and the "subjective" definition of "innovation". Yet, patents are commonly acknowledged as a good "measure" of innovation. In this

direction, Figure 2 illustrates how it was found that the number of patents across the European Union increased in areas demonstrating greater cluster strength/concentration [26].

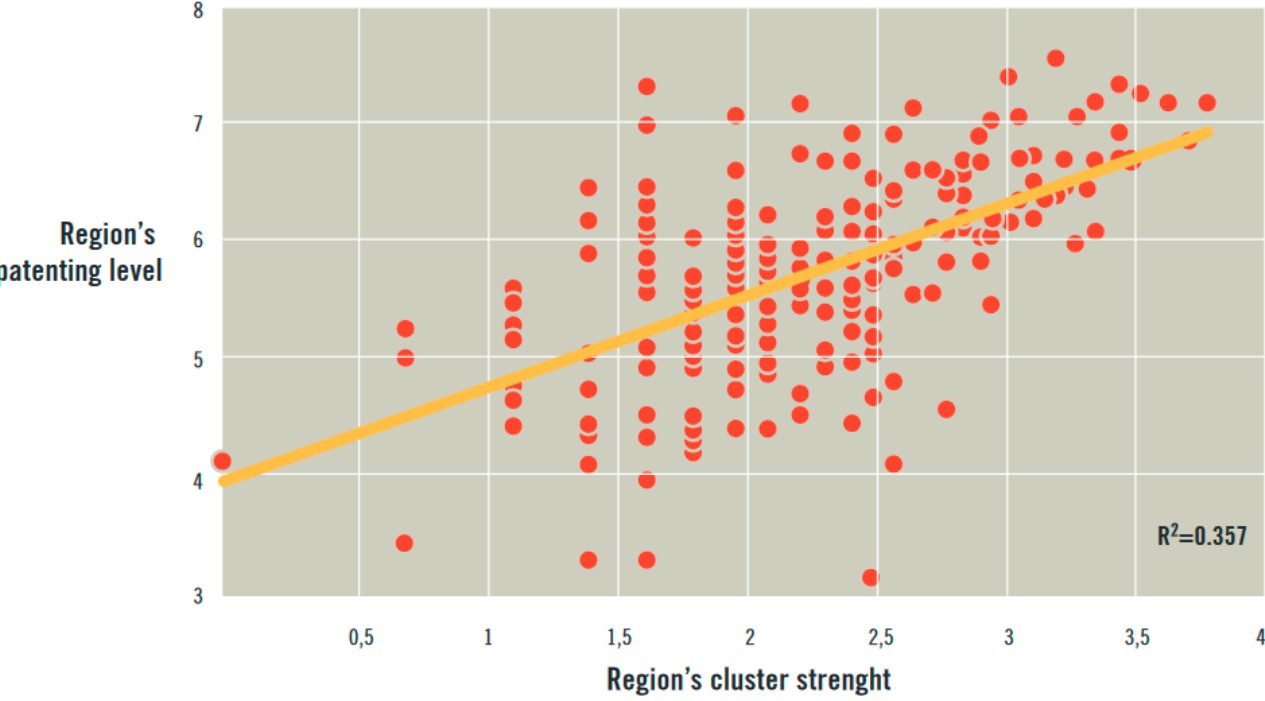

**Figure 2.** Cluster Strength and Patenting Level in European Regions. Source: "The Concept of Clusters and Cluster Policies and their role for Competitiveness and Innovation: Main statistical results and lessons learned".

Given that, one can say that any industrial cluster can be seen as a "hub" [28] where knowledge and associated technologies are circulated/shared. In a cluster initiative, the generation of innovation lies in some cases with the technological "experts" familiar with the dynamics and trends of the specific global markets rather than in the hands of inexperienced executives of smaller firms. This, in turn, aids smaller firms that now have access to more structured and "explicit" knowledge, changing the focus of their business development outlook toward the future rather than just building on past experiences and portfolios [29]. The reason is that the technical experts provide to the area knowledge "sourced" from external resources, behaving as "gatekeepers of knowledge [30]". Companies that have greater financial and institutional resources will facilitate the creation of technology within a cluster in a more coordinated manner [23], using patent data from the industrial district for automatic packaging machinery in Emilia-Romagna, Northern Italy, found that the development and diffusion of new technologies within a cluster are driven by leading or focal firms [30].

This, in some cases, may have adverse effects on creating innovation/knowledge for the cluster as the "innovation building process" depends on a few dominant actors' strategies [31,32]. Thus, one can safely conclude that every company does not participate equally in the different activities of a cluster. On the contrary, companies that have greater financial and institutional resources will facilitate the creation of technology within a cluster in a more coordinated manner [23], proving that in this type of cluster, the collaborative structures are hierarchical and asymmetric [22].

In some cases, innovation and knowledge can be assimilated and integrated by actors not situated within a cluster's geographical locus. Clusters are significant for multinational enterprises (MNEs), which by their very nature are network firms. In some cases, MNEs capture location-specific tacit knowledge created within clusters and serve as "pipelines" between clusters, disseminating knowledge and innovation [33]. In many cases, therefore,

access to innovation does not arise solely from local and regional interaction but is also achieved by creating strategic partnerships between actors of international scope and range [34].

Government: According to policymakers, practitioners, academics, and business leaders, the government's three fundamental roles in the economy are to provide appropriate macroeconomic conditions, enhance microeconomic capacity, and establish a supportive and progressive regulatory framework. Michael Porter contends that while these roles are necessary, they may not be sufficient to ensure successful clustering initiatives. The government should also facilitate and improve cluster development and should also create the proper environment for a productive dialogue bringing cluster participants together. "Cluster development can be enhanced by conscious private and public action," as Porter states [35].

Still, the argument is not if governments can create clusters but if they can provide the business, innovative institutional, and regulatory environments vital for cluster success [36]. Although the debate is still open and vivid [37], one can safely conclude that the key role for government is to enable the creation of clusters. Whether in the form of providing direct access to finance or in less direct ways through creating enabling policy frameworks, strategic action plans, and the provision of trained, motivated public service employees [36].

When it comes to cluster initiatives, it should be noted that, typically these are initiated primarily by governments (32%), then by the industry (27%), or equally by both (35%) [38]. Therefore, government involvement at the stage of cluster initiation accounts for a total of 67% of clusters. More specifically, Sölvell, Lindqvist, and Ketels [38] provide the below-related data:

- In 32% of the cases, the initiative to set up a cluster comes from the government. In 27% of the cases, the initiative comes primarily from the industry, in 5% from universities, while in 35% jointly from two or more parties (usually from government and industry).
- In financing, governments' involvement and contribution are even more critical, as in 54% of the cases, the government is the primary funding source. In comparison, only 18% of clusters are primarily funded by industry, 1% by universities, 2% by international organizations, and 25% by two or more parties.

We should also mention that it is not only the central/national government that facilitates creating a cluster. The local and/or regional governments and/or institutions also play a significant role in this direction. One can mention the endeavors of local and regional governments and institutions of Quebec, Canada, and France, where their facilitating efforts are of significant importance for creating clusters [39].

## 3. Methodology: Using Markusen as a Starting Point and a Methodological Reference

Despite the ongoing debate regarding its limitations, the case study approach is extensively utilized across various scientific disciplines and fields [40]. Using case studies benefits the researcher by offering flexibility, allowing them to explore rather than predict the study output. They are, therefore, free to discover and address issues arising in their field of study. The most challenging aspect of applying this methodological approach is upgrading the research from a descriptive account of "what happens" to a piece of research that can claim to contribute something to the theoretical approach to the topic under discussion [41]. Considering the above, one can argue that the originality of the subject of this article suggests and leads to the adoption of the case study as the appropriate research method. To achieve that, we will use Markusen's typology/distinction of four different types of industrial districts: the Marshallian/Italianate type, the hub-and-spoke, the satellite industrial platforms, and the state-anchored clusters.

Markusen's research provides an extensive analysis of her proposed typology, but this paper only focuses on a summary of its key aspects. The summarized aspects include:

1. The size and number of participating companies within the cluster and their organizational structure.

2. The extent to which companies are integrated within the geographical or institutional entity of the cluster, as well as the connections they have developed within and beyond the region.
3. The management of innovation within the cluster.
4. The presence or absence of a public entity serving as an anchor for the cluster. Additionally, we will try to delineate any inter-European relation the clusters under discussion have by mainly analyzing two different parameters:

- Participation of cluster members in supported international (European) projects/ innovation and/or research programs.
- Cooperation of the studied clusters with other European clusters and with institutions and/or structures of the E.U.

## 4. Clusters: Markusen's Categorization

It is a common belief enhanced and reinforced by a solid and growing body of literature [5–8] that there is not only one type of clusters but several types that have different characteristics. For example, Mytelka and Farinelli [5] make two broad distinctions regarding clusters' classifications. The first is between clusters that originate as spontaneous agglomerations of enterprises and other related actors and those that are induced by public policies. On the other hand, Gordon and McCann [6] distinguish between three models:

- Industrial-complex Model: These industrial complexes are characterized by sets of identifiable and stable relations among firms that are in part manifested in their spatial behavior. The connections are conceived primarily in terms of trading links, and it is these patterns of sales and purchases that are seen as principally governing their locational behavior.
- The Model of Pure Agglomeration: The pure agglomeration model assumes that actors within a cluster operate independently with no cooperation beyond their individual interests in a competitive and atomized environment. The model suggests that profitable local interactions occur through a combination of chance, the law of large numbers (which increases the likelihood of suitable partners being available), and the natural selection of businesses that benefit from the opportunities available.
- The Social-network Model: In this type of cluster, the relationships between the parties of the cluster are built on rules and regulatory norms that essentially cover the totality of the cluster behaviors.

Finally, Markusen's distinctions of industrial clusters are based both on the role of large firms and the state [42] and different interorganizational patterns and arrangements [43]. Markusen's research presents a more diverse picture than the ones mentioned earlier by identifying four distinct types of clusters (Table 1):

1. Marshallian clusters consist mainly of locally owned SMEs [43] and are characterized by significant cooperation levels among these SMEs [42]. Marshallian clusters are also characterized by low degrees of cooperation or linkage with firms external to the district and a high level of "embeddedness" to the district, which creates a unique local cultural identity [8]. The "bonds" created between the companies of the cluster are based on "interactions" that promote trust and a "sense of belonging", reducing transaction costs and facilitating the exchange of information and knowledge through the existence of interpersonal relationships, enhanced by intensive exchanges of personnel between the firms of the cluster [44]. The cluster members create and share innovation [8], while knowledge transfer is both intended and unintended and is often the result of proximity and employees' mobility between companies [45]. Cooperation is formally encouraged by government-sponsored industry organizations [46].
2. Hub-and-spoke types of clusters have one or a few dominant firms surrounded by multiple smaller suppliers [18]. The clusters' structure is hierarchical and unsymmetrical [22]. It is defined by the existence of companies with greater financial and institutional weight, which delineates the development and structure of the clus-

ter [23], acting as a hub. The hub companies are located within the location of the cluster [47]. The importance of the hub companies in the formation and sustainability of a cluster is highlighted by the work of Carbonara [48] (2002), who researched clusters from Italy, concluding that the most dynamic of them modified their configuration and structure. The most prominent of the changes/modifications was the increasingly important role of large firms, with a leading/hub position within the cluster. A well-known example of a district with hub-and-spoke clusters is Seattle, where Boeing acts as the hub for the aerospace industry and Microsoft for the software industry, while the Fred Hutchinson Cancer Center and the University of Washington "shaped" the faith and structure of the local biotechnology industry [18]. Another example of a hub-and-spoke cluster is that of the East Midlands Aerospace cluster in the U.K. The cluster's hub firm is the British engine manufacturer Rolls-Royce, and the spokes are its many second- and third-tier suppliers and other SMEs [49]. The leading firms of the hub-and-spoke clusters act as a "gatekeeper" for the clusters, enabling them to connect with global networks, affecting their sustainability [23], and also "regulating" and shaping the innovation process of the cluster [45]. Under this context, Malipiero, Munari, and Sobrero [30] conclude that hub companies act as "engines of innovation, internally generating new and sophisticated knowledge," and by leveraging on their intellectual and social capital, they act as "technological gatekeepers" facilitating the absorption and internal dissemination of knowledge. Hub companies usually have stronger ties to national trade associations than local, as they tend to lobby more on the national than local level [18].

3. Satellite platform: As in the hub-and-spoke type of clusters, the structure of a satellite platform cluster is somehow hierarchical and unsymmetrical [22], typically consisting of an assemblage/concertation of branch facilities of externally based multinational firms [50,51]. One of the satellite platform clusters that is frequently mentioned in the literature is that of the Research Triangle Park in North Carolina, which groups together several R&D centers of high-tech multinational firms [50,51]. Other examples of satellite platform clusters are the aerospace clusters of Mexico, such as the one situated in Baja California [52,53]. In such types of clusters, the remotely located "parent" company/ies make crucial decisions for the local companies consisting of the core of the cluster, thus "shaping" the structure and potentiality of the cluster [50]. The capabilities and knowledge provided by local companies lead to a form of cooperation between the local aspects of a cluster and externally based multinational firms, creating a "multiple diamond" cluster composition rather than a "single diamond" composition [54]. When it comes to innovation, the multinational "parent" companies are simultaneously a knowledge generator and a knowledge seeker, as Rugman and Verbeke [54] conclude, also playing the role of "global pipelines" diffusing knowledge [55]. Such pipelines are beneficial for the accumulation of knowledge only if the "local aspects/firms" of the cluster are either characterized by a "high-quality local buzz" or are weakly endowed in terms of knowledge as Morrison et al. [55] concluded. The local and/or national government's role is to provide infrastructure, tax breaks, and other generic business inducements [8].

4. State-anchored: While in the types mentioned above of clusters (Marshallian, Hub-and-spoke, Satellite platform) already discussed, the initiative for the creation and the management of them is mainly taken by companies (locally owned SMEs—Marshallian clusters, hub companies—hub-and-spoke clusters, and satellite "parent" companies—satellite platform clusters), in this type of clusters the activity of the member companies is "anchored" to one or several large, governmental institutions such as military bases, state or national capitals, large public universities, etc. [8]. We should not fail to notice that governmental help is provided to all types of clusters. The difference in the state-anchored cluster is, as Markusen and Park [56] concluded in their research on the case of the Changwon cluster, South Korea, the state's role as the lead agent, a factor that lessens the importance of traditional locational aspects.

The Changwon cluster was established because of the state's dedication to building a military supply sector. State-anchored clusters are characterized by centrally coordinated innovation aligned with the public objectives of the anchor institution [20], while the creation of innovation is not significantly dependent on the members of the cluster [45] or on the development of the cluster.

**Table 1.** Markusen's typology of clusters: A synopsis.

| Markusen's Typology of Clusters: A Synopsis | | | | |
|---|---|---|---|---|
| | **Marshallian** | **Hub-and-Spoke** | **Satellite Platform** | **State-Anchored** |
| Characteristics of the cluster's members | Locally owned SMEs | One, or a few, hub firm/s, surrounded by multiple smaller suppliers | Assemblage/concertation of branch facilities of externally based multinational firms | A government-owned or supported entity surrounded by related suppliers (cluster members) |
| Innovation | Members of the cluster create and share innovation | Hub firms "regulating" and shaping the innovation process of the cluster, having the rule of knowledge "gatekeepers" | Multinational "parent" companies are simultaneously a knowledge generator and a knowledge seeker/"global pipelines" and "agents" of knowledge diffusion | Innovation is centrally coordinated, putting any activity in line with the objectives of the "anchor" institution |
| Governmental institutions | Government-sponsored industry organizations | Hub companies have stronger ties to national trade associations than local | Local and/or national government provide infrastructure, tax breaks, and other generic business inducements | Anchor institution/state is the lead agent |
| Cooperation with companies and/or other entities not part of the cluster | Low degrees of linkage with firms external to the district/high level of "embeddedness" to the district, unique local cultural identity | Defined by the hub firm/s | Defined by the "parent" multinational firm/s | Extended with the institution the cluster is "anchored" to |

Source: Authors' own estimations and evaluation.

Belussi [57] argues that the analytical framework mentioned earlier is limited in that it only provides a static snapshot. This means that a cluster can make a transition from one type to another. Markusen provides Detroit as an example, being "transformed" from a Marshallian district to a hub-and-spoke district [8]. Additionally, in the "real world", clusters may have characteristics from different types of Markusen's distinction of industrial clusters. In Italy, for example, the Marshallian clusters are evolving through the consolidation of several leading firms, de facto modifying their configuration and structure to a hub-and-spoke district [48,57].

To investigate our propositions, we will use Markusen's distinction of industrial districts as a framework for analysis, following the lead of numerous other studies on industrial clusters' dynamics. We will proceed with the study of two cases: the French Aerospace Valley cluster of the Midi-Pyrénées and Aquitaine regions and the Andalucia Aerospace cluster [58]. As already set/discussed, the main research question is to investigate and identify in which of the four types of industrial clusters described by Markusen the two clusters belong to.

Markusen's research provides an extensive analysis of her proposed typology, but this paper only focuses on a summary of its key aspects. The summarized aspects include:

1. The size and number of participating companies within the cluster and their organizational structure.
2. The extent to which companies are integrated within the geographical or institutional entity of the cluster, as well as the connections they have developed within and beyond the region.

3.   The management of innovation within the cluster.
4.   The presence or absence of a public entity serving as an anchor for the cluster.

## 5. Case Studies

### 5.1. Analyzing the French Aerospace Valley Cluster

Aerospace Valley is a non-profit organization created in 2005, with the purpose to leverage the competitiveness and visibility of its more than 750 members and to optimize the attractiveness of the Midi-Pyrénées and Aquitaine regions' Aeronautics, Space, and Embedded Systems sectors.

The main industry actors of the Cluster are some of the biggest aerospace and defence companies in France, such as Airbus, Safran, Thales, Dassault, and research institutes such as ONERA, CNES and INRIA [59].

The growth of small and medium-sized enterprises is promoted and cooperation is encouraged between these enterprises, industry leaders, investors, and research organizations to harness the capabilities and assets of the two regions in terms of research, innovation, and know-how. Aerospace Valley is certified as a "Worldwide Cluster". The cluster constitutes a remarkable pool of concentrated aeronautics, space, and embedded system activities, comprising:

- 130,000 industrial jobs;
- 1500 business establishments;
- 1/3 of France's workforce in aeronautics and over 50% in the space sector;
- 8500 researchers;
- 2 of France's 3 prestigious aeronautics and space engineering schools.

Apart from fostering the bonds of its members, the Aerospace Valley cluster also develops extensive links with other European Aerospace clusters, such as the Pegase Cluster, the European Aerospace Cluster Partnership (EACP), and the Network of European Regions Using Space Technologies (NEREUS).

At the institutional level, Aerospace Valley has managed national innovation programs since its foundation. Since 2005, a total of 1327 research projects were submitted, of which, 598 were funded. Financing amounted to EUR 1.517 billion, of which EUR 885 million came from private funding while the remaining EUR 632 million was from governmental funds [59]. In addition, the cluster has participated in 53 projects funded by the E.U. (in 16 of these projects, the cluster has the role of the coordinator) [60].

The cluster favors collective innovation projects between co-localized partners by profiting from the positive effects of proximity. The significant domination and control of the big players of the cluster characterize these projects [61,62].

As is clearly understood from the above, the Aerospace Valley cluster has managed to create an appropriate "environment" for the region's development by concentrating a considerable amount of resources (human, technology, infrastructure, etc.). Some 1500 business establishments with specialized expertise in the related fields are located in the geographical area. This reflects the main aim of the cluster, which can be summarized as the concentration of as many resources as possible "under one roof" facilitating the concept of "mega clusters" [63].

In order to address "concrete" technological challenges, or to focus work in a particular product/service direction, mega clusters typically establish "focus areas". For example, the Aerospace Valley cluster has established nine "Strategic Positioning Communities", as they refer to them), to function as a "think tank" for defining and driving the cluster strategy, fostering innovative partnership projects and leveraging the potential of each member and project within the cluster. These "Communities" are overseen by a team of renowned experts in the particular field. The nine Strategic Positioning Communities of the Aerospace Valley cluster specifically consist of:

- Structures, Materials, and Processes;
- Energy and Electro-mechanical Systems;

- Air Transport Safety and Security;
- Navigation Telecommunications and Observation;
- Embedded Systems, Electronics, and Software;
- Man–Machine Interface;
- Maintenance, Repair, and Overhaul;
- Future Factory;
- Highly complex system design and integration.

The Aerospace Valley cluster has established alliances with several other concerted initiatives in "complementary" directions (such as Optitec (optics—photonics), SCS (information technology), Mer PACA (technologies of the sea), Capenergies (future energies) and Risques (risk management)). Below in Table 2, one can assess the main characteristics of the Aerospace Valley, according to Markusen's typology.

**Table 2.** Main characteristics of the Aerospace Valley, according to Markusen's typology.

| Aerospace Valley | | | |
|---|---|---|---|
| **Members** | **Innovation** | **Governmental Institutions** | **Collaboration with External Structures** |
| SMEs and large companies | Creating strategic synergies between academia and the business sector | Manages research projects funded by the French state | Developing extensive links with other French and European aerospace clusters |

Source: Authors' own estimations and evaluation.

To summarize, one can say that the Aerospace Valley cluster has achieved the following:

- Creating strategic synergies between academia and the business sector: The Aerospace Valley cluster has created "strategic synergies" bringing together the business sector and academia enabling the creation of a strong innovation base, by enhancing access to new knowledge. It is worth repeating that two of France's three most prestigious aeronautics and space engineering schools are members of the cluster. Enabling interpersonal relationships between the students of the schools and the companies has led to the creation of a "technological circle", mostly based on the so-called "tacit knowledge", the knowledge that is difficult to circulate, mainly due to the nature of the message it conveys and which is better diffused through face-to-face interactions.
- Creating jobs for highly qualified personnel: It is worth mentioning that the cluster mostly provides jobs for qualified personnel, as 1/3 of France's aeronautics professionals and over 50% of those in the space sector work for companies that are members of the Aerospace Valley cluster.
- Forging strategic collaborations with other clusters: The Aerospace Valley cluster from its initial steps forged strategic collaborations/alliances with other European space clusters. The cluster has also formed partnerships with clusters from other industrial domains, such as agriculture and ICT. Since the applications/technologies developed by the cluster's companies can be used in several industrial fields, the cluster can diversify its market reach.

## 5.2. Analyzing the Andalusian Aerospace Cluster

The creation of an industrial cluster offers a region several benefits and opportunities, as it brings together the main economic actors, namely the industry, government, and educational sector, and provides the appropriate infrastructure for all of them to work cooperatively [64–66]. The region's economic resources (private and public) are used in a more structured and rationalized way, thus achieving a more constructive return on investment [64,67–69]. The case of the Andalusian Aerospace cluster highlights the benefits a region can have from the economic and business synergies a cluster creates.

The Andalucía Aerospace cluster is a private association created to represent all the aerospace companies of Andalusia. All 71 cluster members belong to the private sector. The main goals of the cluster are:

- To promote a sustainable scientific and technological development of the Andalusian aerospace and industrial sector;
- To contribute actively to the training and education of professionals in the sector;
- To promote business excellence through synergies between partner companies;
- To encourage and facilitate interactions between member companies in the aerospace industry;
- To act as institutional representation vis-à-vis public institutions and organizations at the national and international levels.

The total turnover of the cluster members is over 2.4 billion euros, representing 72% of the turnover of the Andalusian aerospace supply chain, while exports in 2021 reached 1.1 billion euros. Moreover, cluster members employ approximately 14,500 people, representing 65% of the Andalusian aerospace supply chain employment, whereas the cluster has experienced a rapid growth of 190% in the last decade.

The main reason behind this rapid cluster development is Airbus's decision to set up its third final assembly line (FAL) in Seville [70]. Specifically, the beginning of the A400M aircraft mass production generated a work volume for the Andalusian cluster, valued at more than EUR 130 million, and it is expected that within a 20-year period, after the delivery of orders already agreed on by seven European countries, the program will generate a turnover of EUR 4.4 billion for the cluster's participants. Fifteen (15) companies of the Hélice cluster have received direct orders in relation to the A400M program. Further to this, it is estimated that more than 63% of the Andalusian auxiliary industry has directly or indirectly benefited from works related to the A400M program [71].

The decision made by Airbus was influenced in part by the Andalusian government's (Junta) strong commitment, demonstrated through its Strategic Aeronautics Industry Plan, which has instilled stability and confidence in the sector's companies. Additionally, the government has facilitated the development of scientific and technological parks specifically tailored to the aerospace industry: Aerópolis, Tecnobahía & PTA, CATEC, CFA, and ATLAS [72]. The importance of Airbus's subcontracting for this Andalusian cluster is also highlighted in the chart below. Sales to Airbus represent 86% of the total sales of the cluster, while sales to other companies such as Boeing and Bombardier are significantly lower [73].

It is worth mentioning that it is not only the leading companies of the cluster that contributed to this. On the contrary, the "auxiliary" companies of the cluster have also vastly contributed toward this direction, as more than 63% have directly or indirectly benefited from works related to the A400M program.

Finally, it is essential to highlight that apart from the obvious economic benefits the cluster brings to the region, it also creates a solid innovation base, by enhancing access to new knowledge, through participating in substantial R&D programs such as the following:

- The goal of the ASSETs + project is to establish a sustainable supply chain of human resources and redefine the required skill sets in the military sector.
- AERIS aims to enhance the competitiveness of companies in the aeronautical sector in the Andalucía-Alentejo cross-border region by facilitating innovation and technology transfer.
- The objective of the prestigious project is to enhance collaboration among UAV clusters in Europe and assist SMEs in their international expansion.

The Hélice Andalusian Aerospace Cluster also provides SMEs with access to R&D&I activities and innovative infrastructure that they would not have accessed in the absence of the cluster [74]. Without the existence of the clusters' structures, large companies would work with these SMEs in a standard "client–supplier" logic rather than associate with them as partners and allow them to anticipate technological breakthroughs and entrepreneurial synergies [14]. Therefore, most of these SMEs would not be able to work in these highly challenging technical areas, or at least they would not be able to enhance their cooperation synergies with the large companies of the cluster.

One should also highlight Airbus's importance in the structure of the cluster and in the diffusion of knowledge. Several authors and studies have analyzed the "bond"

between multinational companies and clusters [75]. Such studies have largely proven that clusters are essential for multinational companies, which can "integrate" and diffuse their knowledge in clusters. Multinational companies can also use the knowledge created within the clusters and, in addition, play the role of "knowledge pipelines" between two or more clusters and/or local entities [33]. Therefore, access to innovation does not result exclusively from local and regional interaction but is also obtained through strategic partnerships with international actors [34]. In the table below (Table 3), one can preview the main characteristics of the Andalusian Aerospace Valley, according to Markusen's typology.

**Table 3.** Main characteristics of the Andalusian Aerospace Valley, according to Markusen's typology.

| Andalusian Aerospace Cluster | | | |
|---|---|---|---|
| **Members** | **Innovation** | **Governmental Institutions** | **Collaboration with External Structures** |
| SMEs and large companies | Innovation is diffused to "auxiliary" companies by leading companies | Andalusian government has a Strategic Aeronautics Industry Plan<br><br>Promoted the construction of scientific and technological parks specializing in the aerospace industry | Airbus plays a dominant role: Sales to airbus represents 86% of the total sales of the cluster |

Source: Authors' own estimations and evaluation.

## 6. Conclusion and Discussion

The France Aerospace Valley characteristics resemble those of a hub-and-spoke cluster. The cluster has managed to create a solid innovation base, with supporting R&D activities, thus generating a competitive advantage by enhancing access to new knowledge and boosting a creative output for the members. Since 2005, cluster members submitted, through collaborative efforts, a total of 1327 research projects [59]. In addition, the cluster has participated in 53 projects funded by the E.U. (in 16 of these projects, the cluster has the role of the coordinator) [60]. The cluster favors collective innovation projects between co-localized partners by profiting from the positive effects of proximity. The significant domination and control of the big players of the cluster characterize these projects [61,62].

Overall, one can safely conclude that companies acting in a "synchronized" manner/direction (such as is the case in the Aerospace Valley cluster) are more dynamic and competitive, mainly because they "share" their knowledge and they built upon this enlarged "pool" of knowledge capital. This process of "collective efficiency" [76] is adopted by the participating companies and, hence, the cluster as a whole. In turn, this leads to the more rapid evolution of the respective clusters and the creation of new businesses, as the knowledge acquired/shared will eventually lead to enhanced production processes, higher-quality products/services, etc., ultimately leading to economic growth.

The "collective efficiency" accompanied and augmented by proximity and the dominance of the big player of the cluster creates an environment where companies with greater financial and institutional weight facilitate the creation of technology in a more coordinated manner [23], tending though to control this process [30]. The innovation created within the cluster is then distributed through solid structures to other external actors, as the cluster has created extensive links with French and European clusters. On the other hand, the Andalucia Aerospace cluster's characteristics resemble those of a satellite platform cluster. The structural development of the cluster and the relations between its members are somehow hierarchical and unsymmetrical [22], and they are based mainly on Airbus subcontracting activity, which is reinforced by the fact that Airbus represents 86% of the total sales of the cluster, while sales to other companies such as Boeing and Bombardier are significantly lower (6% and 2%, respectively) [73].

When it comes to the creation of innovation, Airbus can be simultaneously characterized as a knowledge generator and a knowledge seeker [54], also playing the role of a "global pipeline" diffusing knowledge [55]. Such pipelines are beneficial for the accumulation of knowledge only if the "local aspects/firms" of the cluster are either characterized

by a "high-quality local buzz" or are weakly endowed in terms of knowledge [55], exemplifying what Aguilera and Guerrero [58] describe as a collaborative paradigm. The "bond" between Airbus and the cluster helped the cluster to experience rapid growth, reaching 190% in the last decade. The total turnover of the cluster members represents 72% of the turnover of the Andalusian aerospace supply chain, while cluster members employ approximately 14,500 people, representing 65% of the Andalusian aerospace supply chain employment. The growth in the aerospace industry has been made possible with the help of the Andalusian government, which has implemented a tangible Strategic Aeronautics Industry Plan and encouraged the development of scientific and technological parks focused on the aerospace sector [72].

Additionally, one can safely conclude that both clusters have a European Aspect (please, also see Table 4 below, where the characteristics of the clusters under discussion are presented). The Aerospace Valley cluster has established alliances with several other concerted initiatives in "complementary" directions "disseminating" the knowledge created to other parts of Europe. The Andalusian cluster "assimilates" technology from other parts of Europe and distributes it to its members. This means that clusters have the role of disseminating the knowledge, acting in a "synchronized" manner/direction, mainly since they "share" their knowledge and build upon this enlarged "pool" of knowledge capital. In turn, this leads to the more rapid evolution of the respective technologies and the creation of new knowledge as the experience acquired/shared eventually leads to enhanced processes of knowledge generation and the structural optimization of the involved entities at a rather European and not only local level.

**Table 4.** Characteristics of the clusters under discussion.

| Cluster | Members | Innovation | Governmental Institutions | Collaboration with External Structures |
|---|---|---|---|---|
| Andalusian Aerospace cluster | SMEs and large companies | Innovation is diffused to "auxiliary" companies by leading companies | Andalusian government has a Strategic Aeronautics Industry Plan<br><br>Promoted the construction of scientific and technological parks specializing in the aerospace industry | Airbus plays a dominant role: sales to airbus represent 86% of the total sales of the cluster |
| French Aerospace Valley cluster | SMEs and large companies | Creating strategic synergies between academia and the business sector | Manages research projects funded by the French state | Developing extensive links with other French and European aerospace clusters |

Source: Authors' own estimations and evaluation.

It would also be essential to emphasize that both clusters are closely related to E.U. R&D funding initiatives. They position themselves as a platform of project identification and preparation for available funding and support programs at the E.U. level, creating and integrating them into an extensive "European network of clusters" and other entities [77]. Through this process, the clusters increase their cross-border cooperation and promote cross-border diversification and the creation of synergies.

By analyzing these two clusters, the paper provides insights into the unique features and challenges of industrial clusters in Europe and the opportunities for cross-border collaboration and knowledge sharing. Moreover, the paper highlights the importance of European Union (E.U.) policies and initiatives in promoting the development of industrial clusters and innovation ecosystems. It argues that E.U. policies, such as the Horizon 2020 research and innovation program, can facilitate cross-border collaboration and knowledge exchange and provide financial support for the development of innovative projects and initiatives.

This comparative analysis provides policymakers and industry leaders with valuable insights into the factors that contribute to the success and competitiveness of industrial clusters in different contexts. It also highlights the role of policy interventions and E.U.

financial and institutional frameworks in developing industrial clusters in Europe, particularly in fostering innovation and knowledge creation, supporting the development of small and medium-sized enterprises, and promoting collaboration between industry, academia, and government. Last but not least, it provides a better understanding of the development of industrial clusters in Europe and offers policy implications for their further growth and competitiveness by referring to a pan-European system of actors in which clusters are essential members.

**Author Contributions:** Conceptualization, V.K.; methodology, V.K.; formal analysis, V.K.; investigation, V.K.; resources, V.K.; writing—original draft preparation, V.K. and T.M.; writing—review and editing, V.K. and T.M.; supervision, T.M. All authors have read and agreed to the published version of the manuscript.

**Funding:** This research received no external funding.

**Conflicts of Interest:** The authors declare no conflict of interest.

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
