# Peer review of "Markusen’s Typology with a “European” Twist, the Examples of the French Aerospace Valley Cluster and the Andalucia Aerospace Cluster"

_world, doi:10.3390/world4010013_

Round 1

Reviewer 1 Report

The positive aspects of the work.

The scientific article reveals important aspects of the classification of economic and geographical processes using the clustering method. This topic is becoming relevant all over the world in the context of changing commodity and financial flows. The authors substantiate the importance of studying the problem with a significant number of approaches to clustering in industries in the absence of consistency. Therefore, the study is of interest to both the professional and scientific community, as well as to a wide range of readers. It should be noted the well-structured structure of the work, conducting an extensive review of the literature, which is confirmed by the list of sources. Both descriptive and analytical methods were used. The material is presented in an accessible form, the illustrations are clear. The conclusions are thorough and economically justified.

Remarks.

1. The scientific novelty of the research is not defined in the article. This topic has been studied quite a lot in modern science, so you need to justify what the author's contribution is.

2. Already at the beginning of the study, we are talking about clusters, but the definition of this concept is not given. It is necessary to clarify this concept from the point of view of the author's approach, since clusters are used in different spheres and interpreted differently.

3. In the introduction, it is necessary to define the purpose of the study in order to understand whether the result has been achieved.

Conclusion.

The article, of course, is characterized by scientific novelty and practical significance, and can be recommended for publication after the elimination of comments or reasoned explanations of the authors regarding the issues raised.

Author Response

Dear Reviewer, please see ur response

REVIEWER 1 - RESPONSE

  1. The scientific novelty of the research is not defined in the article. This topic has been studied quite a lot in modern science, so you need to justify what the author's contribution is.

Answer: The scientific novelty of the research lies in its "European" twist, which refers to the articles' focus on the specificities of inter-European collaboration between industrial clusters. By analyzing these two aerospace and defence clusters in France and Spain, the paper provides insights into the unique features and challenges of industrial clusters in Europe and the opportunities for cross-border collaboration and knowledge sharing.

Moreover, the article highlights the importance of European Union (EU) policies and initiatives in promoting the development of industrial clusters and innovation ecosystems. It argues that EU policies, such as the Horizon 2020 research and innovation program, can facilitate cross-border collaboration and knowledge exchange and provide financial support for developing innovative projects and initiatives, creating synergies and strategic alliances between different actors of the EU clusters' ecosystem.

Finally, the study aims to highlight the role of policy interventions in the development of industrial clusters in Europe, particularly in fostering innovation and knowledge creation, supporting the development of small and medium-sized enterprises, and promoting collaboration between industry, academia, and government. Overall, the study aims to provide insights into the unique features and challenges of industrial clusters in Europe and the opportunities for cross-border collaboration and knowledge-sharing.

  1. Already at the beginning of the study, we are talking about clusters, but the definition of this concept is not given. It is necessary to clarify this concept from the point of view of the author's approach, since clusters are used in different spheres and interpreted differently.

Answer: The definition of a cluster in this study is not limited to the physical proximity of firms and institutions but also considers the nature and intensity of their interactions and relationships and the specific characteristics of their industry. Therefore, the definition goes beyond the traditional concept of the physical proximity of firms and institutions. While physical proximity is an essential factor that enables face-to-face interactions and knowledge exchange, the definition of a cluster in this study also considers the nature and intensity of firms' interactions and relationships.

Therefore, the definition provides a nuanced and multidimensional concept that considers the nature and intensity of the interactions and relationships between firms and institutions, the specific characteristics of their industry or field, as well as the EU institutional and policy framework that supports and shapes their development. This multidimensional concept of a cluster allows for a more comprehensive analysis of the factors that contribute to the success and competitiveness of industrial clusters and provides policymakers and industry leaders with a more nuanced understanding of the strategies and policies needed to support their growth and innovation.

  1. In the introduction, it is necessary to define the purpose of the study in order to understand whether the result has been achieved.

Answer: The purpose of this study, is to investigate in which of the four types of industrial clusters described by Markusen, the French Aerospace Valley cluster of the Midi-Pyrénées & Aquitaine regions and the Andalucia Aerospace cluster belongs to. Adding to Markusen's typology, we will also try to delineate these two clusters' "European Aspects". We will examine if they have developed any "inter-European" synergy/ies with other entities (clusters, companies, EU institutions etc.) of the EU ecosystem. The study aims to provide insights and policy implications for policymakers and industry leaders interested in supporting their growth and competitiveness.

Overall, the study aims to contribute to the ongoing conversation on how to support the growth and competitiveness of industrial clusters in Europe and beyond, and to provide policymakers and industry leaders with valuable insights and policy implications for achieving this goal, by also using EU “tools” and financial frameworks.

Reviewer 2 Report

The study is theoretical and contains two interesting case studies. I present my comments below.

1.   The purpose of the article should not be to investigate. We study a problem for a purpose, not for the sake of research.

2.   What new does this article bring to science? The paper needs to be stronger in this respect. A large part of the article is devoted to the description of cluster types, according to Markunsen’s. The essence of the article is the description of two case studies. It's weak for a scientific paper.

3.   Figures 1 and 2 are scans of drawings. The content of Figure 2 is practically illegible.

4.   The data in Figures 1, 2, and 3 are almost 20 years old. There are double titles in Figures 1 and 3.

5.   There needs to be a header in table 2. Tables 3 and 4 have only one row each. Figure 4 does not specify the source and from which year these data come.

Author Response

Dear reviewer, please see our response

REVIEWER 2 - RESPONSE

The study is theoretical and contains two interesting case studies. I present my comments below.

  1. The purpose of the article should not be to investigate. We study a problem for a purpose, not for the sake of research.

Answer: The purpose of this study, is to investigate which of the four types of industrial clusters described by Markusen, the French Aerospace Valley cluster of the Midi-Pyrénées & Aquitaine regions and the Andalucia Aerospace cluster belongs to. Adding to Markusen's typology, we will also try to delineate these two clusters' "European Aspects". We will examine if they have developed any "inter-European" synergy/ies with other entities (clusters, companies, EU institutions etc.) of the EU ecosystem. The study aims to provide insights and policy implications for policymakers and industry leaders interested in supporting their growth and competitiveness.

Overall, the study aims to contribute to the ongoing conversation on how to support the growth and competitiveness of industrial clusters in Europe and beyond and to provide policymakers and industry leaders with valuable insights and policy implications for achieving this goal by also using EU “tools” and financial frameworks.

  1. What new does this article bring to science? The paper needs to be stronger in this respect. A large part of the article is devoted to the description of cluster types, according to Markunsen's. The essence of the article is the description of two case studies. It's weak for a scientific paper.

The scientific novelty of the research lies in its "European" twist, which refers to the articles' focus on the specificities of inter-European collaboration between industrial clusters. By analyzing these two aerospace and defence clusters in France and Spain, the paper provides insights into the unique features and challenges of industrial clusters in Europe and the opportunities for cross-border collaboration and knowledge sharing.

Moreover, the article highlights the importance of European Union (EU) policies and initiatives in promoting the development of industrial clusters and innovation ecosystems. It argues that EU policies, such as the Horizon 2020 research and innovation program, can facilitate cross-border collaboration and knowledge exchange and provide financial support for developing innovative projects and initiatives, creating synergies and strategic alliances between different actors of the EU clusters' ecosystem.

Finally, the study aims to highlight the role of policy interventions in the development of industrial clusters in Europe, particularly in fostering innovation and knowledge creation, supporting the development of small and medium-sized enterprises, and promoting collaboration between industry, academia, and government. Overall, the study aims to provide insights into the unique features and challenges of industrial clusters in Europe and the opportunities for cross-border collaboration and knowledge-sharing.

  1. Figures 1 and 2 are scans of drawings. The content of Figure 2 is practically illegible.

Answer: I have updated the figure. I hope that it is now illegible.

  1. The data in Figures 1, 2, and 3 are almost 20 years old. There are double titles in Figures 1 and 3.

Answer: The data is indeed old. But, it is essential to note that while the data in Figures 1, 2, and 3 may be almost 20 years old, the analysis and insights presented in the study are still relevant and valuable for policymakers and industry leaders.

Answer: I erased the title from figures 1 and 3.

  1. There needs to be a header in table 2. Tables 3 and 4 have only one row each. Figure 4 does not specify the source and from which year these data come.

Answer: Table 2 (please refer to the picture below) is part of table 1.

As requested, I also add the source and the year the data comes.